# *Momordica cochinchinensis* Aril Ameliorates Diet-Induced Metabolic Dysfunction and Non-Alcoholic Fatty Liver by Modulating Gut Microbiota

**DOI:** 10.3390/ijms22052640

**Published:** 2021-03-05

**Authors:** Hsiu-Chen Huang, Chiung-Ju Chen, Yu-Heng Lai, Yu-Chun Lin, Wei-Chung Chiou, Hsu-Feng Lu, Ying-Fang Chen, Yu-Hsin Chen, Cheng Huang

**Affiliations:** 1Center for Teacher Education, National Tsing Hua University, Hsinchu 30041, Taiwan; jane@mail.nd.nthu.edu.tw; 2Department of Applied Science, Nanda Campus, National Tsing Hua University, Hsinchu 30041, Taiwan; 3Department of Medical Technology, Jen-Teh Junior College of Medicine, Nursing and Management, Miaoli 35042, Taiwan; T002913@ms.skh.org.tw; 4Department of Pathology and Laboratory Medicine, Shin Kong Wu Ho-Su Memorial Hospital, Taipei 111, Taiwan; 5Department of Chemistry, Chinese Culture University, Taipei 11114, Taiwan; lyh21@ulive.pccu.edu.tw; 6Institute of Medical Device and Imaging, National Taiwan University, College of Medicine, Taipei 10617, Taiwan; ylinz@umich.edu; 7Department of Biotechnology and Laboratory Science in Medicine, National Yang Ming Chiao Tung University, Taipei City 112, Taiwan; ryanw.chiou@gmail.com; 8Departments of Clinical Pathology, Cheng Hsin General Hospital, Taiwan 112, Taiwan; ch1835@chgh.org.tw; 9Department of Restaurant, Hotel and Institutional Management, Fu-Jen Catholic University, Taipei 24205, Taiwan; 10Taitung District Agricultural Research and Extension Station, Council of Agriculture, Taitung City 95059, Taiwan; yfchen@mail.ttdares.gov.tw; 11Taichung District Agricultural Research and Extension Station, Council of Agriculture, Changhua 515, Taiwan; ychen@tdais.gov.tw

**Keywords:** adiposity, gut microbiota, insulin resistance, *Momordica cochinchinensis* aril, nonalcoholic fatty liver

## Abstract

Obesity and its associated conditions, such as type 2 diabetes mellitus (T2DM) and nonalcoholic fatty liver disease (NAFLD), are a particular worldwide health problem at present. *Momordica cochinchinensis* (MC) is consumed widely in Southeast Asia. However, whether it has functional effects on fat-induced metabolic syndrome remains unclear. This study was conducted to examine the prevention effect of *Momordica cochinchinensis* aril (MCA) on obesity, non-alcoholic fatty liver and insulin resistance in mice. MCA protected the mice against high-fat diet (HFD)-induced body weight gain, hyperlipidemia and hyperglycemia, compared with mice that were not treated. MCA inhibited the expansion of adipose tissue and adipocyte hypertrophy. In addition, the insulin sensitivity-associated index that evaluates insulin function was also significantly restored. MCA also regulated the secretion of adipokines in HFD-induced obese mice. Moreover, hepatic fat accumulation and liver damage were reduced, which suggested that fatty liver was prevented by MCA. Furthermore, MCA supplementation suppressed hepatic lipid accumulation by activation of the AMP-activated protein kinase (AMPK) and peroxisome proliferator-activated receptor-alpha (PPAR-alpha) signaling pathway in the human fatty liver HuS-E/2 cell model. Our data indicate that MCA altered the microbial contents of the gut and modulated microbial dysbiosis in the host, and consequently is involved in the prevention of HFD-induced adiposity, insulin resistance and non-alcoholic fatty liver disease.

## 1. Introduction

As a leading public health issue, obesity has been shown to be associated with various chronic diseases and metabolic disorder, such as cardiovascular diseases, hypertension, type 2 diabetes, non-alcoholic fatty liver disease (NAFLD), and various cancers [1,2]. Studies have shown that a high fat diet may be a risk factor for the development of obesity and metabolic diseases [3]. The liver is one of the major organs involved in fat metabolism. Disturbance of lipid metabolism results in excess accumulation of lipids in the liver, known as fatty liver or steatosis. NAFLD is a form of chronic fatty liver disease that is linked to diet and obesity. NAFLD may cause hepatic damage and progression to other chronic conditions, in which steatosis progresses to non-alcoholic steatohepatitis (NASH), liver fibrosis, cirrhosis, and hepatocellular carcinoma [4,5].

*Momordica cochinchinensis* (MC) is a tropical fruit that is grown predominantly in Southeast Asia. MC is a traditional food and herbal medicine in Vietnam and China. MC is called “gac” in Vietnamese. Its seed is called “Mubiezi” in Mandarin Chinese [6] and used as a Chinese traditional medicine for a variety of purposes, such as detoxification and de-swelling. Previous studies demonstrated that a water extract from MC had potential antitumor activities, by inhibiting human colon tumor cell growth and angiogenesis [7]. In addition, an MC seed extract regulated Bcl-associated apoptosis in a breast cancer cell line, suppressing tumor progression [8]. High levels of carotenoids, such as β-carotene, lycopene and lutein, are expressed in the peel, pulp, aril and seeds of MC [9]. *Momordica cochinchinensis* aril (MCA), also known as seed pulp, has more than eight times the levels of lycopene and β-carotene that are found in tomatoes and carrots [10]. Thus, MCA is commonly used in the daily diet for its rich nutrition [11]. MCA also contains abundant α-tocopherol, essential fatty acids, and flavonoids [12]. A number of publications have focused on the anti-oxidant activity of MC; however, no studies have reported the effect of MCA on metabolic syndromes, dietary fat-induced NAFLD and gut microbiota.

In this study, we examined the effects of MCA on high-fat diet (HFD)-induced obesity, metabolic dysregulations, and NAFLD in vivo. In our mouse model, HFD mice showed abnormalities of lipid and glucose metabolism, as well as significant dyslipidemia and markers of hepatic steatosis. MCA prevented weight gain, changes in lipid and carbohydrate metabolism, and NAFLD in the HFD-induced obese mice. In addition, a human fatty liver cell model, HuS-E/2 immortalized human primary hepatocytes, which we established previously [13], was used to investigate the molecular mechanism through which MCA prevents fatty liver disease. Our findings also provide evidence for the use of MCA for the management of dietary fat-induced adiposity, metabolic syndrome, NAFLD and gut microbiota dysbiosis.

## 2. Results

### 2.1. Effects of Momordica cochinchinensis Aril (MCA) on Body Weight and Food Efficiency in High-Fat Diet (HFD)-Fed Mice

It has been shown that obesity is highly associated with metabolic dysregulation [1]. To determine the effect of MCA on obesity in vivo, five-week-old male C57BL/6J mice were fed a normal diet (ND) or HFD, without or with 1% or 3% MCA for 10 weeks. The weights of HFD mice were significantly higher than those of ND and HFD-MCA mice after 10 weeks of diet (Figure 1A). Interestingly, HFD-3% MCA mice had significantly greater food uptake than the other groups (Figure 1B). The food efficiency ratios (FER) of the HFD-MCA groups were significantly lower than the HFD group (Figure 1C). It is suggested that HFD mice had greater food efficiency than MCA mice, which may contribute to the weight gains in HFD mice.

### 2.2. MCA Decreased Visceral Fat Deposition and Prevented HFD-Induced Hyperlipidemia

Excessive lipid accumulation and visceral fat deposition in the trunk region are features of metabolic syndrome [14]. Therefore, epididymis adipose tissues (EAT) of the mice, after 10 weeks on the different diets, were dissected and measured. We observed that the diameters of the adipose cells of the HFD group were significantly larger than the other groups, and the MCA-fed group showed lower cell diameters and sizes than the adipocytes from the HFD group (Figure 2A,C). The masses of the EAT in the HFD group were significantly greater than the other groups (Figure 2B). The HFD-induced increase in the mass of the EAT was ameliorated by MCA treatment.

Augmented lipid composition in plasma is another well-known characteristic of metabolic dysregulation [15]. Therefore, the effects of MCA on lipemia syndrome were measured in our HFD mouse model to evaluate the degree of metabolic conditions. We found that plasma triglyceride (TG), total cholesterol (TC), and low-density lipoprotein-cholesterol (LDL-C) were significantly higher in the HFD group (Figure 2D,E,G), suggesting that the presence of hypertriglyceridemia and high cholesterol phenomena in HFD mice, consistent with the symptoms observed in obese and diabetic patients. After 10 weeks of treatment, plasma TG were significantly lower in the HFD-3% MCA group, and TC and LDL-C levels were significantly lower in both the HFD-1% and 3% MCA groups (Figure 2D,E,G). No significant differences of plasma high-density lipoprotein-cholesterol (HDL-C) levels were revealed between the groups (Figure 2F). The findings indicate the inhibitory effects of MCA in hyperlipidemia bioactivities.

### 2.3. Effects of MCA on HFD-Induced Insulin Resistance and Plasma Adipocytokines and Glucose-Dependent Insulinotropic Polypeptide (GIP)

Dietary fat intake has been shown to be associated with deteriorated insulin function [16]. Therefore, several indices of insulin resistance symptom were measured in HFD mice, supplemented with MCA. As shown in Figure 3A, the fasting blood glucose levels of the HFD mice were significantly higher than the ND group at week 10. Levels of high fasting blood glucose were normal in the HFD-MCA groups. We also found that the insulin level was lower in HFD-3% MCA mice (Figure 3B). Moreover, the area under the curve (AUC) of glucose levels was calculated using the intraperitoneal glucose tolerance test (IPGTT) assay, which is an index of glucose tolerance. Increased glucose tolerance was found in the HFD group and mice that were fed an HFD supplemented with MCA had significantly lower glucose tolerance than the mice fed an HFD alone (Figure 3C). Insulin resistance is commonly calculated using the homeostasis model assessment of insulin resistance (HOMA-IR) [17]. It is shown that treatment with 3% MCA was able to maintain normal HOMA-IR levels in HFD mice (Figure 3D). These data suggest MCA prevented the HFD-induced abnormality of glucose homeostasis and insulin sensitivity.

Furthermore, leptin and resistin, two important adipocytokines that are closely associated with glucose homeostasis insulin sensitivity [18], were measured. The levels of plasma leptin and resistin were significantly higher in the HFD mice than the ND mice and the HFD-MCA mice had lower plasma leptin and resistin levels than the HFD mice (Figure 3E,F). In addition, glucose-dependent insulinotropic polypeptide (GIP) has been reported to stimulate insulin secretion in a glucose-dependent manner [19]. Levels of plasma GIP were also found to be much lower in the HFD-MCA mice than the HFD mice (Figure 3G). These results suggest MCA might contribute to glucose homeostasis and insulin sensitivity via regulation of those adipocytokines and GIP.

### 2.4. MCA Treatment Reduced HFD-Induced Fatty Liver and Inhibited Liver Damage

A high-fat diet has been shown to promote metabolic syndromes, such as non-alcoholic fatty liver disease, characterized by triglyceride accumulation in hepatocytes [20]. To monitor the degrees of lipid deposition in liver, we weighed the livers and measured the hepatic triglyceride (TG) and hepatic cholesterol levels of the mice. The liver weights of the HFD group were significantly greater than the ND group, and the HFD-MCA groups had lower liver weights than the HFD group (Figure 4A). The hepatic TG levels were significantly lower in the HFD-MCA groups than the HFD group (Figure 4B). Moreover, the hepatic cholesterol levels of the HFD group were higher than the ND and HFD-3% MCA groups (Figure 4C). The liver tissues of different groups were then dissected and examined. The HFD mice had swelling tissue with a foaming morphology of the hepatocytes (Figure 4D). MCA treated groups had much lesser lipid deposition in the hepatocytes than the HFD group.

It has been also shown that HFD may cause hepatic steatosis and changes in the balance of β-oxidation and oxidants, which in turn affects body weight, insulin signaling and other metabolic parameters [21]. Levels of glutamic oxaloacetic transaminase (GOT) and glutamic pyruvic transaminase (GPT) in the plasma of the mice were measured as markers of hepatic lipotoxicity and as an indication of steatohepatitis [22]. We found the levels of plasma GOT and plasma GPT were upregulated in HFD mice and this increase was prevented significantly by treatment with 3% MCA (Figure 4E,F). Nevertheless, the levels of the renal disease marker creatinine (CRE) and blood urea nitrogen (BUN), and pancreatitis marker lipase (LIP) did not differ significantly among the groups (Figure 4G–I). These results suggest that MCA treatment successfully reduced fatty liver and the severity of HFD-induced liver damage and MCA was not harmful to the kidney and pancreas.

### 2.5. MCA Treatment Downregulates Lipid Accumulation and AMP-Activated Protein Kinase (AMPK)/ACC Phosphorylation, and Upregulates Peroxisome Proliferator-Activated Receptor-Alpha (PPAR-α) Activity in a Human Fatty Liver Cell Model

In-depth investigation of the effects of MCA on fatty acid deposition in liver cells, we used immortalized human primary HuS-E/2 hepatocytes as a human fatty liver cell model [13]. HuS-E/2 cells were incubated with MCA and the MTT (3-(4,5-dimethylthiazol-2-yl)-2,5-diphenyltetrazolium bromide) assay was used to evaluate the viability of the cells. We found that MCA concentrations of 10–100 µg/mL had no significant effect on cell viability (Figure 5A). Therefore, MCA concentrations below 100 µg/mL were used in subsequent studies. To induce cellular lipid accumulation, HuS-E/2 cells were incubated in media containing 0.1 mM of oleic acid (OA) and intracellular lipid accumulation was measured using oil red O staining. As shown in Figure 5B,C, treatment with 50 and 100 µg/mL of MCA significantly reduced OA-induced cellular lipid accumulation in HuS-E/2 cells.

The AMP-activated protein kinase (AMPK) has been suggested to play a crucial role in regulating fat metabolism in the liver [23]. As shown in Figure 5D, 50 µg/mL and 100 µg/mL MCA significantly increased AMPK phosphorylation in OA-treated HuS-E/2 cells. Activation of AMPK’s downstream target enzyme, ACC, by phosphorylation at Ser-79 (pACC) was also assessed. Treatment with 10 to 100 µg/mL of MCA increased ACC phosphorylation in OA-treated HuS-E/2 cells. In addition, peroxisome proliferator-activated receptor-alpha (PPAR-alpha) is expressed predominantly in the liver, heart and kidneys, and its activation promotes utilization and catabolism of fatty acids [24]. As shown in Figure 5E, treatment with 100 µg/mL MCA significantly increased PPAR-alpha activity in OA-treated HuS-E/2 cells. Taken together, MCA facilitated AMPK and ACC activation and stimulated lipid metabolism through PPAR-alpha activation in cells under high fat conditions.

### 2.6. MCA Consumption Modulates Obesity-Driven Dysbiosis of the Gut Microbiota

There is growing evidence that diet contributes to obesity-associated changes in the gut microbiota [25,26]. Studies have suggested that gut microbiota is altered by obesity and plays a critical role in the development of diabetes [27]. To investigate whether MCA affects gut microbiota of the HFD mice, we collected faecal samples from the various study groups and performed an Illumina sequencing analysis of bacterial 16S rRNA. Operational taxonomic unit (OTU)-based principal coordinates analysis (PCoA) revealed distinct clustering of the microbiota compositions for each different group (Figure 6A). Taxonomic profiling showed that the treatment of MCA led to a decrease in the Firmicutes to Bacteroidetes ratio in HFD-fed mice, although the value remained higher than those of the ND group (Figure 6B). Because the Firmicutes and Bacteroidetes are the two hallmarks of obesity-driven dysbiosis, the findings implied that MCA prevented microbial dysbiosis.

An unweighted UniFrac tree of the resulting set of 16S rRNA gene sequences demonstrated several clusters based on bacterial community membership (Figure 6C, upper panel). This indicated significantly separate microbiota between the ND and HFD groups as well as HFD and 1% MCA groups. A hierarchical clustering heatmap of the top 26 ranked OTUs showed that HFD-fed mice had a distinct microbial profile, compared with ND mice. Moreover, the relative abundances of 26 OTUs that were altered by MCA and the changing direction of represented bacterial taxa information (species level) modulated by MCA are shown (Figure 6C, lower panel). The heatmap data revealed that treatment of MCA altered the microbial composition, which was different from that of the HFD-fed mice. Then, the linear discriminant analysis (LDA) effect size (LEfSe) for the most discriminating OTUs of the different groups was calculated to explore the taxa within the lowest taxonomic level possible. The mean abundance of 29 OTUs differed significantly between the HFD and ND groups and a total of 16 OTUs was more abundant in the ND group (Figure 6D, upper panel). The s_*bifidum*, f_*Bifidobacteriaceae*, and g_*Bifidobacterium* were discriminating faecal bacterial communities of the MCA group that differed from the HFD group (Figure 6D, lower panel). Taken together, MCA treatment was effective at changing in gut microbial populations in HFD-fed mice.

## 3. Discussion

*Momordica cochinchinensis* (MC) has been consumed as a traditional food and its seed used in Chinese medicine. MC is reported to be rich in phytochemicals, including carotenoids and flavonoids [28]. Products of MC/Gac were released into the market in the forms of powder, oil capsules, juice, frozen fruit and others as food additives or to serve for medicinal uses [29]. The beneficial activities of the bioactive compounds in MC were found to have the ability to scavenge free radicals [29]. The MC aril (MCA) is more rich in lycopene and β-carotene than tomatoes and carrots [10]. The high levels of nutritionally important compounds that are found in MC rejuvenated the cultivation of MC and may promote the related food and pharmacological industries.

Excess weight and obesity-related disorders have become a global epidemic [30]. Visceral obesity has been shown to be correlated with the development of chronic inflammation and metabolic complications, such as type 2 diabetes, hepatic steatosis and cardiovascular diseases [14]. In this study, we established a mouse model of metabolic syndrome by feeding the mice with an HFD and tested the effects of MCA on these mice. HFD mice showed greater food efficiency ratios (FERs) than ND mice and administration of MCA significantly prevented the increased FERs in HFD mice (Figure 1). HFD mice also had higher lipid accumulation in the trunk region and increased diameters of the adipocytes than ND mice. Adding MCA supplements to the HFD prevented increased visceral fat deposition and adipocyte size in the HFD mice (Figure 2). Although the body weight of 3% MCA group is not significantly different from that of 1% MCA group, 3% MCA demonstrates a more pronounced effect regarding other pathophysiological indices, including fat weight, HOMA-IR and liver weight, compared with 1% MCA. These findings suggest that MCA may play a crucial role in preventing visceral obesity.

It also has been shown that obesity is associated with insulin resistance [2]. Insulin resistance is also linked to increased numbers of adipocytes, inflammation, and oxidative stress, which contributes to the progression of NAFLD [31]. We found the level of blood glucose was significantly higher in the HFD group and supplementation with MCA prevented increases in the glucose tolerance levels and HOMA-IR levels of the HFD-MCA groups (Figure 3). These findings suggest MCA prevented increases in the level of glucose in plasma, and improved insulin sensitivity. Lycopene has been shown to have significant inhibitory effects on HFD-induced insulin resistance by preventing the expression and phosphorylation of STAT3 in a mouse model [32]. In addition, plasma ß-carotene has been shown to have reversed correlation with insulin resistance [33]. MCA, that contains high amount of carotenoids, may ameliorate metabolic syndrome and improve insulin sensitivity, and provide an alternative dietary strategy to overcome metabolic syndrome.

It has been reported that patients with chronic liver diseases have low antioxidant levels in liver tissue and serum [34]. Carotenoid levels in the plasma were significantly lower in patients with NASH than in control subjects [35]. Therefore, carotenoids and other antioxidant micronutrient levels might be associated with the development of obesity and steatohepatitis [36,37]. MCA was reported to contain high amount of carotenoids, such as lycopene and β-carotene and lutein [9]. It is likely that these antioxidant micronutrients in the MCA contribute to the preventive effects against hepatic steatosis.

A recent study showed adipose tissue is not only a mediator of systemic lipid storage, but an endocrine organ that secretes hormones known as adipokines, such as resistin and leptin [38]. Dysregulation of adipokine production is related to ectopic fat accumulation and insulin resistance. Our data showed the levels of resistin and leptin were increased in mice that were fed an HFD. Treatment of MCA significantly prevented the secretion of both adipokines and GIP (Figure 3). These data imply that MCA decreased HFD-induced lipid accumulation by inhibiting adipogenesis and regulating adipokine secretion.

Various studies have shown that lean individuals have a greater microbial diversity than obese individuals [25,27]. Obesity-related disorders have been suggested that the functionality and microbial complexity of the gastrointestinal tract contribute to local and systemic health [25,27]. In our study, Illumina sequencing analysis of bacterial 16S rRNA in the faeces of the mice revealed that MCA treatment altered the gut microbiota of the HFD-fed mice. We inferred that the beneficial effects of MCA may be attributed to the metabolism of lycopene, β-carotene and lutein by gut microbiota, and lead to an alteration of microbial composition. The results of our study suggest that MCA might modulate lipid accumulation and weight-loss by maintaining the diversity of gut microbiota.

Several studies have shown that obese individuals appear to have increased levels of Gram-positive *Firmicutes* phylum over Gram-negative *Bacteroidetes* phylum [39,40]. We observed that the ratio of *Firmicutes* to *Bacteroidetes* (F:B) ratio was higher in HFD-fed mice, and supplementation with MCA was able to reverse to a normal F:B ratio. This observation suggests MCA treatment results in population growth of important bacteria inferred to be protective or beneficial. Interestingly, our data showed that the relative abundance of the family *Bifidobacteriaceae* and *Bifidobacterium bifidum* was higher in the MCA group than the HFD group. Previous studies showed that the probiotics containing *Bifidobacterium bifidum* play an important role in preventing and treating patients with obesity and insulin resistance [41,42]. Our data showed that MCA treatment increase *Bifidobacterium bifidum* in HFD-fed mice (Figure 6D), suggesting a protective role against the development of NAFLD and obesity. It was also shown that obese patients with non-alcoholic steatohepatitis have a higher abundance of *Lactobacillus* spp. than healthy individuals [43]. These previous results are in agreement with our findings that HFD-fed mice had symptoms of steatohepatitis and a higher abundance of *Lactobacillus* spp. (Figure 6D). Other studies showed that bifidobacteria antagonize Enterobacteriaceae and *Enterococcus* and thus might protect the infants from enteropathogenic infections [44,45]. Therefore, prevention of microbial dysbiosis and its associated diseases by MCA might be through an increase in the populations of beneficial species, such as Bifidobacteriaceae.

## 4. Materials and Methods

### 4.1. Preparation of Lyophilized Momordica cochinchinensis Aril (MCA)

The habitat of Taiwan’s native *Momordica cochinchinensis* locates in the area of Hualien County and Taitung County in Taiwan. Taitung District Agricultural Research and Executive Station (Taitung County, Taiwan) collected and planted the composite strain of Taiwan’s native *Momordica cochinchinensis* originated from the wild for research investigation. *Momordica cochinchinensis* was divided carefully into its anatomical components and *Momordica cochinchinensis* aril (MCA) was thoroughly homogenized using a homogenizer (D-500, WIGGENS, Straubenhardt, Germany) and stored in a freezer (−18 °C ± 2 °C) until dehydration. The frozen MCA sample was lyophilized at −40 °C for 72 h using a floor type lyophilizer (FD12, KINGMECH Co., Tu-Cherng City, Taiwan). Then, the lyophilized MCA was stored at −20 °C as functional materials. *Momodica cochinchinensis* aril was extracted with n-hexane/methanol/acetone. Lycopene and β-carotene standards (Sigma-Aldrich, St. Louis, MO, USA) were prepared with acetone. Samples were analyzed using high-performance liquid chromatography (HPLC) to analyze lycopene and β-carotene, respectively. The amount of lycopene was determined by the intensity at the wavelength of 475 nm at the retention time between 9–10 min. β-Carotene was determined by the intensity at the wavelength of 450 nm at the retention time of 37.4 ± 2.5 min. The lyophilized MCA contains nutrients including 0.82 ± 0.05 mg of lycopene and 1.78 ± 0.04 mg of β-carotene per gram.

### 4.2. Animals

Thirty-two 4-week-old male C57BL/6J mice were purchased from BioLASCO Taiwan Co, Ltd. All mice were individually housed under a constant temperature (24 °C) and 12 h light/dark cycle at the Animal Center of the National Yang-Ming University, Taipei, Taiwan. They were housed with four per cage and had free access to food and drinking water. Mice fed a standard diet and adapted to the environment for one week were subsequently divided randomly into four groups and fed a normal diet (ND, n = 8), high-fat diet (HFD, n = 8, 30% fat and 1% cholesterol), or HFD with 1 or 3% (weight for weight) lyophilized MCA (each group, n = 8) for 10 weeks. At the end of the experimental period, all mice were sacrificed. Plasma samples, liver tissue, and epididymis adipose tissue were harvested for further analysis. The use of animals for this research was approved by the Animal Research Committee of the National Yang-Ming University (IACUC no.107-0213) and all procedures followed The Guide for the Care and Use of Laboratory Animals (NIH publication, 85-23, revised 1996) and the guidelines of the Animal Welfare Act, Taiwan. ND (Cat. 5010) and HFD (Cat. 58V8; 494 kcal per 100 g, 45% energy from fat) were purchased from LabDiet Inc. (St. Louis, MI, USA) and TestDiet Inc. (St. Louis, MI, USA), respectively.

### 4.3. Morphology of the Liver and Fat Tissues

The liver and epididymal adipose tissue were removed from each mouse. Samples were subsequently fixed in 10% paraformaldehyde/phosphate-buffered saline (PBS) and embedded in paraffin for staining with Hematoxylin and eosin. All the specimens were examined under a microscope (Axiolab 5, Carl Zeiss Inc., Oberkochen 73447, Germany) at 200× magnification.

### 4.4. Biochemical Analysis of Plasma

The total plasma triglyceride (TG), total cholesterol (TC), high-density lipoprotein-cholesterol (HDL-C), glutamic oxaloacetic transaminase (GOT), glutamic pyruvic transaminase (GPT) levels were measured using enzymatic assay kits by FUJI DRI-CHEM analyzer (Fujifilm). Resistin, leptin, glucose-dependent insulinotropic polypeptide (GIP) levels were measured by multiplex assay (Yu Shing Bio-Tech Co., Taipei City, Taiwan). The low-density lipoprotein-cholesterol (LDL-C) level was calculated as [(TC) − (HDL-C) − (TG/5)].

### 4.5. Blood Glucose, Plasma Insulin, and the Homeostasis Model Assessment of Insulin Resistance Index

Every 2 weeks, the 12 h fasting blood glucose was measured in tail vein blood with a glucose analyzer (EASYTOUCH, Miaoli County, Taiwan). Enzymatic assay was used to measure the plasma insulin concentration (Insulin Ultra-Sensitive Assay kit HTRF^®^, Cisbio, Bedford, MA, USA). Intraperitoneal glucose tolerance tests (IPGTTs) were performed in all mice 10 weeks after the start of the study. Mice fasted for 16 h were injected intraperitoneally with glucose 1.0 g/kg body weight and the blood glucose levels were measured in tail vain blood at 0, 30, 60, 90, 120 and 150 min. The homeostasis model assessment of insulin resistance (HOMA-IR) was calculated as [fasting insulin concentration (mU/L) × fasting glucose concentration (mg/dL) × 0.05551]/22.5.

### 4.6. Triglyceride and Cholesterol Analysis of Liver Tissue

For triglyceride and cholesterol determinations, mouse liver tissues were extracted and analyzed using triglyceride and cholesterol quantitation assay kits (Abcam, Cambridge CB2 02X, UK), respectively, according to the manufacturer’s instruction.

### 4.7. Antibodies and Reagents

Antibodies against AMPK, pACC (Ser 79), ACC and tubulin were obtained from Genetex, the anti-pAMPK (Thr 172) antibodies were from Millipore, and the horseradish peroxidase-conjugated anti-mouse and anti-rabbit IgG antibodies were from Jackson ImmunoResearch Laboratories Inc., West Grove, PA, USA. Oil red O was purchased from Sigma-Aldrich, USA.

### 4.8. Cell Culture

HuS-E/2 cells, kindly provided by Dr. Shimotohno (Kyoto University, Japan), were maintained as described previously in primary hepatocyte medium (PH medium) [46]. To simulate the fatty liver disease model, HuS-E/2 cells at 70% confluence were incubated with 0.1 mM oleic acid (OA) for 18 h. To measure the intracellular lipid content, HuS-E/2 cells were stained using the Oil Red O as described previously [13].

### 4.9. Cell Viability Assay

Cell viability was assessed using the 3-(4,5-dimethylthiazol-2-yl)-2,5-diphenyltetrazolium bromide (MTT) assay. MTT assays were performed as described previously [13].

### 4.10. Western Blot Analysis

After treatment, HuS-E/2 cells were harvested in lysis buffer (50 mM Tris-HCl, pH 8.0, 5 mM EDTA, 150 mM NaCl, 0.5% Nonidet P-40, 0.5 mM phenylmethylsulfonyl fluoride, and 0.5 mM dithiothreitol). The protein concentrations of the supernatants were determined using a protein assay kit (Bio-Rad 94547, Hercules, CA, USA), then equal amounts of total cellular protein (100 mg) were resolved by sodium dodecyl sulfate polyacrylamide gel electrophoresis (SDS-PAGE), transferred onto polyvinylidene difluoride membranes (Amersham Biosciences, Piscataway, NJ 08855, USA), and probed with primary antibody, followed by horseradish peroxidase-conjugated secondary antibody, then bound antibody was visualized using enhanced chemiluminescence kits (Amersham Biosciences, Piscataway, NJ 08855, USA).

### 4.11. PPAR-Alpha Activity Assay

The activity of human PPAR-alpha was measured by enzyme-linked immunosorbent assay (ELISA) using protocols supplied by the manufacturer (Abcam, Cambridge CB2 02X, UK).

### 4.12. Gut Microbiota Analysis

The fecal samples were collected 3 days before the end of experimental period. Fecal genomic DNA was extracted using a QIAamp DNA Stool Mini Kit (Qiagen, Dusseldorf, Germany) according to the manufacturer’s instructions. The 16S library was performed according to the library preparation guide for Illumina MiSeq System (part # 15044223 Rev. B). The V3F/V4R primers (V3F: 5′-CCTACGGGNGGCWGCAG-3′/V4R: 5′-ACTACHVGGGTATCTAATCC-3′) for the hypervariable region of the 16S rRNA gene with overhang sequence were used for metagenome analysis to generate Illumina 16S library by two-step polymerase chain reaction (PCR) from genomic DNA. The first stage PCR for amplifying V3-V4 region was performed in duplicate for each DNA sample. The two PCR products of each sample were pooled and subjected to the second PCR using a Nextera XT DNA index kit to add multiplexing indices and Illumina sequencing adapters, according to the 16S metagenomic sequencing library preparation guide. The 16S libraries were pooled and sequenced on a MiSeq with MiSeq V3 reagent paired 300-bp reads. The QIIME 2 software package (version 2018.8) [47] was used to process the raw sequence data. The sequences were dereplicated, quality filtered and chimera removed with q2-dada2 [48]. Representative sequence sets for each dada2 sequence variant were used for taxonomic classification. OTUs were clustered by scikit-learn naive Bayes machine-learning classifier and assigned against the curated Greengenes v13.8 reference database at the QIIME2 website [49]. Microbial diversity was visualized using principal coordinate analysis (PCoA) of Unweighted UniFrac distances. The mean of relative abundance in each group was compared at the phylum, family, and genus levels. Heatmap to display the relative abundance of the most abundant OTUs was generated using Java Treeview (v1.1.6r4) [50].

### 4.13. Statistical Analysis

All values are expressed as the mean ± SEM from at least three separate experiments. GraphPad Prism 6.01 software (GraphPad, San Diego, CA, USA) was used to analyze the experimental data. AUC analysis was performed using the trapezoidal method. One-way analysis of variance (ANOVA) followed by Dunnett’s multiple comparison test was used to compare differences among groups of samples. Asterisks indicated that the values were significantly different from the control (*, *p* < 0.05; **, *p* < 0.01; ***, *p* < 0.001.).

## 5. Conclusions

In summary, we demonstrated that MCA attenuated diet-induced obesity in mice, prevented insulin resistance, and exerted protective effects against visceral fat deposition and lipid accumulation in the liver. The reversed phenomena on metabolism were examined by concentrations of glucose, lipids, adipokines, and other insulin resistance-associated indices. Supplementation of MCA modulated the host microbial composition. These solid results indicate that MCA has promising bioactivity in regulating diet-induced obesity, insulin sensitivity, hyperlipidemia and hepatic steatosis by supporting healthy populations or abundances of inferred important bacteria.

## Figures and Tables

**Figure 1 ijms-22-02640-f001:**
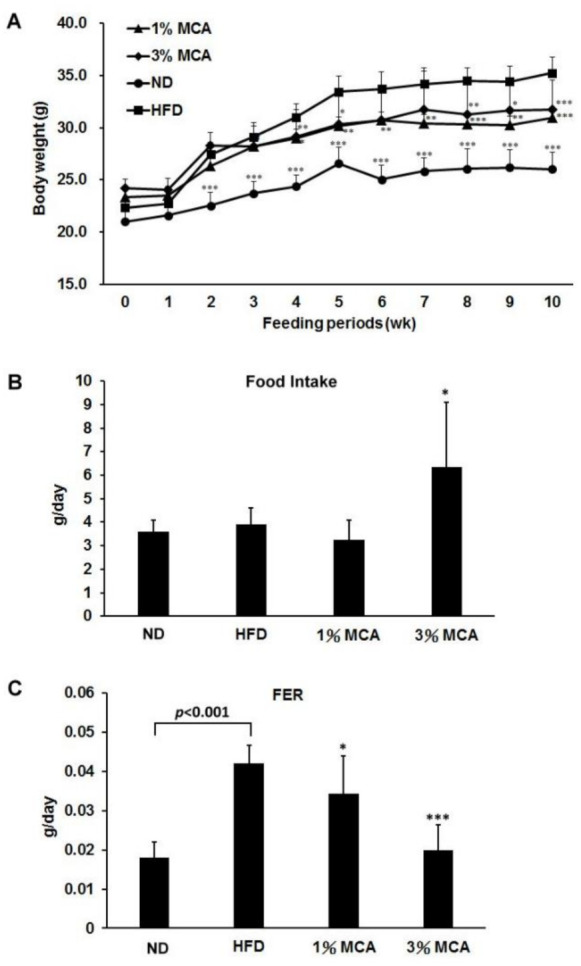
The effect of *Momordica Cochinchinensis* Aril (MCA) on body weights and food intake in high-fat diet (HFD)-fed mice. (**A**) Changes in body weight. (**B**) Food intake. (**C**) Food efficiency ratio (FER). Data are shown as means ± standard error of the mean (SEM). HFD vs. MCA, * *p* < 0.05; ** *p* < 0.01; *** *p* < 0.001.

**Figure 2 ijms-22-02640-f002:**
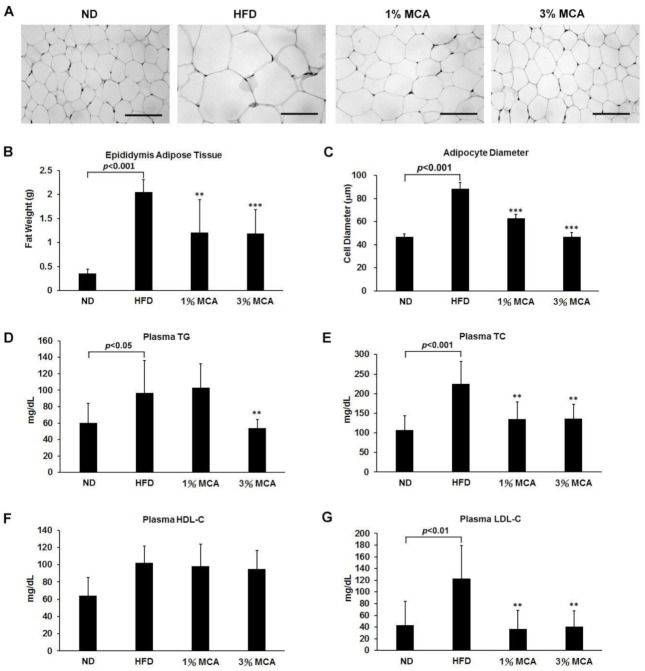
The effect of MCA on fat deposition and plasma lipid levels in HFD-fed mice. (**A**) Hematoxylin and eosin staining showed adipocytes in epididymis adipose tissues (EAT). (**B**) The weight of EAT. (**C**) The adipocytes diameters. (**D**) Plasma triglyceride (TG). (**E**) Total cholesterol (TC). (**F**) high-density lipoprotein-cholesterol (HDL-C). (**G**) Low-density lipoprotein-cholesterol (LDL-C). The scale bar is 100 µm. Data are shown as means ± SEM. HFD vs. MCA, ** *p* < 0.01; *** *p* < 0.001.

**Figure 3 ijms-22-02640-f003:**
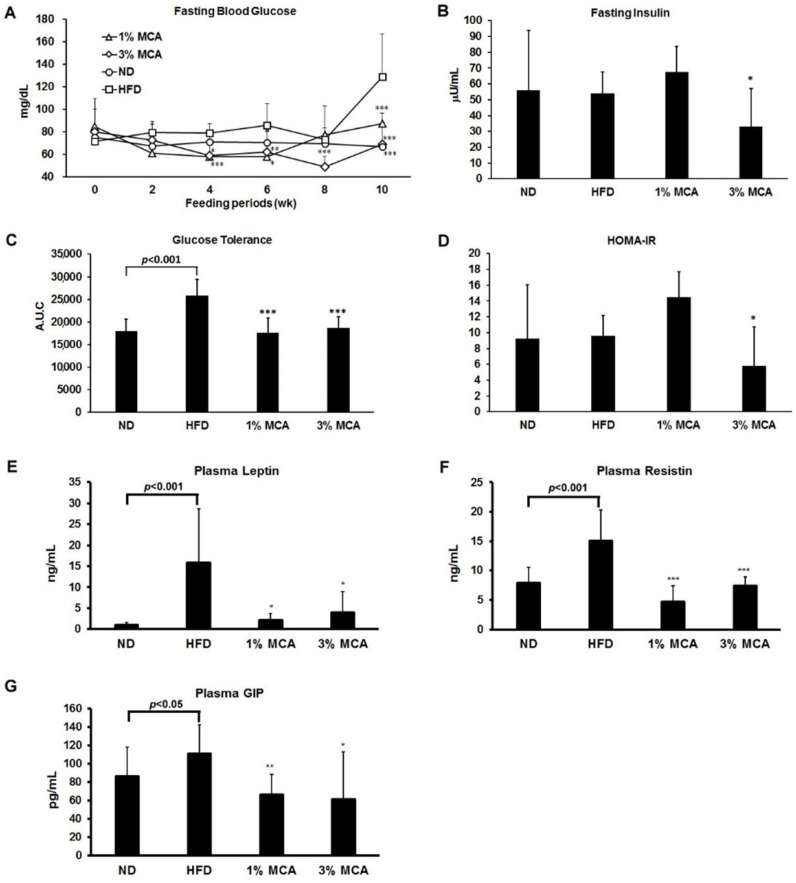
The effect of MCA on insulin resistance and adipocytokines and glucose-dependent insulinotropic polypeptide (GIP) in HFD-fed mice. (**A**) Fasting blood glucose level changes during 10 weeks of MCA treatment. (**B**) Plasma insulin levels after 12 h of fasting. (**C**) Area under the curve (AUC) of blood glucose and insulin levels. (**D**) The homeostasis model assessment of insulin resistance (HOMA-IR) index calculated using fasting blood glucose and insulin levels. Changes in the levels of plasma adipocytokines, leptin (**E**), and resistin (**F**), and GIP (**G**). Data are shown as means ± SEM. HFD vs. MCA, * *p* < 0.05; ** *p* < 0.01; *** *p* < 0.001.

**Figure 4 ijms-22-02640-f004:**
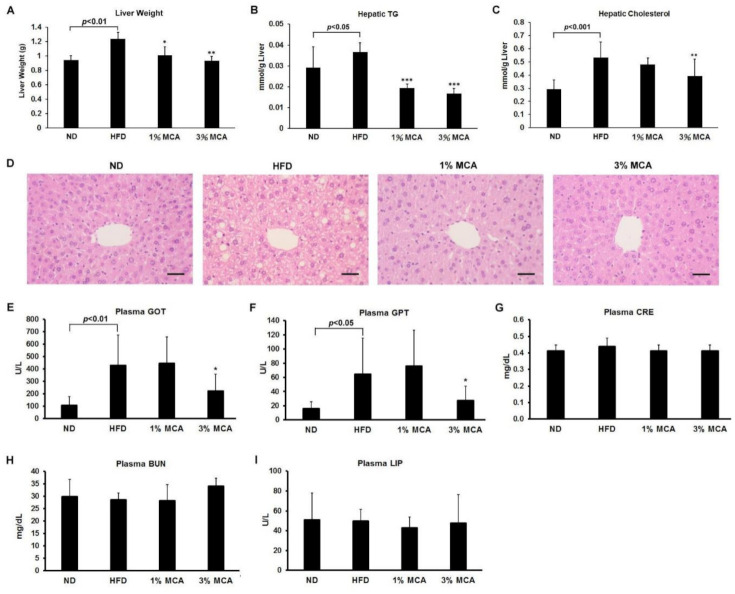
The effect of MCA on lipid accumulation in the liver and hepatic steatosis-related markers in HFD-fed mice. (**A**) Liver weight, (**B**) hepatic TG level, (**C**) hepatic cholesterol, (**D**) hematoxylin and eosin staining of hepatocytes (original magnification ×200), (**E**) glutamic oxaloacetic transaminase (GOT), (**F**) glutamic pyruvic transaminase (GPT). The plasma levels of the kidney lipotoxicity markers (**G**) creatinine (CRE), and (**H**) blood urea nitrogen (BUN). (**I**) The plasma levels of the pancreas lipotoxicity marker lipase (LIP). The scale bar is 100 µm. Data are shown as means ± SEM. HFD vs. MCA, * *p* < 0.05; ** *p* < 0.01; *** *p* < 0.001.

**Figure 5 ijms-22-02640-f005:**
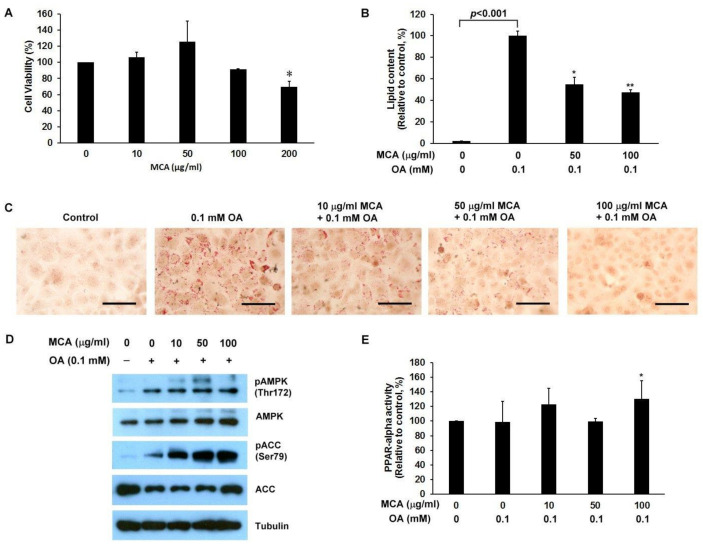
The effect of MCA on oleic acid-induced cellular lipid accumulation in HuS-E/2 cells. (**A**) Cell viability following treatment with MCA for 24 h in HuS-E/2 cells. (**B**) Quantitative analysis of lipid deposition in the Oil Red O-stained HuS-E/2 cells. (**C**) Micrographs of the Oil Red O staining using a microscope at 200× original magnification. Control represents cells without oleic acid (OA) or MCA treatment. (**D**) Western blot analysis for pAMPK and pACC, total AMP-activated protein kinase (AMPK) and ACC. Tubulin served as a loading control. (**E**) The levels of peroxisome proliferator-activated receptor-alpha (PPAR-alpha) activity were examined by enzyme-linked immunosorbent assay (ELISA). The scale bar is 100 µm. Experiments were performed in triplicate and data are presented as mean ± SEM of three independent experiments. Control vs. MCA, * *p* < 0.05; ** *p* < 0.01.

**Figure 6 ijms-22-02640-f006:**
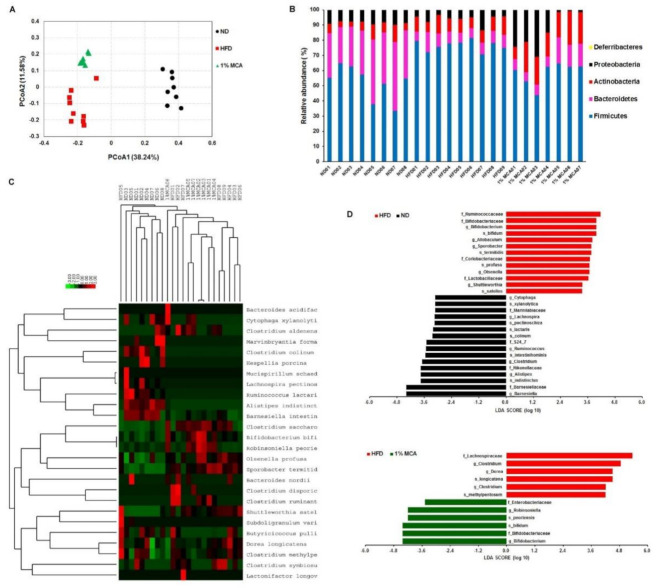
MCA modulated the composition of the HFD-disrupted gut microbiota. (**A**) Principal coordinate analysis (PCoA) of gut microbiota based on operational taxonomic unit (OTU) abundance. (**B**) Bacterial taxonomic profiling at the phylum level of gut microbiota. (**C**) Unweighted UniFrac tree comparing 16S rRNA clone library sequences from the gut microbiotas of different groups (upper panel). The relative abundance of bacterial classes observed in these data sets is represented in heatmaps below each tree (lower panel). (**D**) Linear discrimination analysis (LDA) effect size (LEfSe) was calculated to explore the taxa within the lowest taxonomic level possible that more strongly discriminate between the gut microbiota of ND vs. HFD (upper panel) and 1% MCA vs. HFD (lower panel).

## Data Availability

Not applicable.

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
