# Peer review of "Momordica cochinchinensis Aril Ameliorates Diet-Induced Metabolic Dysfunction and Non-Alcoholic Fatty Liver by Modulating Gut Microbiota"

_ijms, 2021, doi:10.3390/ijms22052640_

Round 1

Reviewer 1 Report

Research Summary:

The authors’ goal was to explore the effects on obesity, metabolic disfunction, and NAFLD when introducing MCA fruit into the diet of mice. They used 24 mice (although indicate 30 initially) in 3 test groups, “Normal Diet”, “High Fat Diet”, and “High Fat Diet with 1% or 3% MCA” although numbers of 1% and 3% mice are not delineated clearly. The author’s looked at liver morphology (sacrifice post-study), TG, TC, HDL-C, GOT, GPT, GIP, and calculated LDL-C levels throughout the study. Human cell culture was additionally used to simulate fatty liver disease with measurements of lipid content. Western blot was used for protein analysis and ELISA used for PPAR-alpha activity. Gut microbiome was performed using 16S amplicon metagenome sequencing with Illumina MiSeq from fecal samples. Sequencing was processed and analyzed using the QIIME2 pipeline. Statistical analysis was performed using standard analysis although the environment and details are not well explained. The author’s conclude that inclusion of MCA with a high fat diet attenuates diet-induced obesity and prevents insulin resistance. They also conclude that there is an MCA induced change in gut microbiota and that gut microbiota homeostasis is maintained through the introduction of MCA to the diet.

General perception:

The author’s did a lot of admirable and interesting work as part of this study. In particular, the biometrics collected e.g. blood metabolite levels are well thought out. I greatly appreciate the parallel investigation of mice with human cell culture to make the findings from the mouse model more human relevant. The delineation between MCA 1% and MCA 3% is not clear throughout the paper and needs additional clarification. I do feel that some of the work could be better presented and that claims are sometimes questionably represented or supported by the data as outlined below.

Detailed comments and questions on sections:

General:

  • Double check comma usage throughout.
  • General proofreading including for things such as proper tense usage (e.g. is vs has and are vs. were 86-57), wording (e.g. “treatment of MCA” vs. “treatment with MCA” 305) and improper sentences (e.g. 166-168).

Figures 2a and 4d:

I suggest adding scale bars to the microscopy images.

Abstract:

  • The claim is that the study was conducted to measure the preventative and therapeutic value of MCA. However, MCA was introduced at the beginning of the study so does not illustrate how therapeutic it is when introduced to already presenting high fat diet phenotypes as a therapy. Perhaps rephrase.

Introduction:

No major comments

Materials and Methods:

347-355           It is written that 30 mice were purchased but based on groupings and n of 8, 24 mice were actually used in the study although this number is never expressly communicated. I suggest making this clearer in the writing.

352-354           Why was no MCA control group established? (ND with MCA 1% and 3%). This would have helped explore MCA gut communities related to MCA.

352-354           Nutritional information should be provided for the diets, whether custom prepared for the study, commercial feeds, or through citation.

353-354           How many mice had 1% MCA and how many mice had 3% MCA? This delineation should be clearly communicated and those numbers should be your true n values.

414      When were fecal samples collected? This is not outlined anywhere in methods. Was the starting microbial community assessed or just the final community? These factors can have implications for findings.

Please see related recent MDPI publication:

LeBrun, Erick S., Meghali Nighot, Viszwapriya Dharmaprakash, Anand Kumar, Chien-Chi Lo, Patrick SG Chain, and Thomas Y. Ma. "The Gut Microbiome and Alcoholic Liver Disease: Ethanol Consumption Drives Consistent and Reproducible Alteration in Gut Microbiota in Mice." Life 11, no. 1 (2021): 7.

418-422           PCR protocol (temperatures, times, polymerase used etc.) should be explained, referenced to a kit, or cited so that it is reproducible.

425      What do you mean when you say that sequences were “replicated” in q2-dada2?

429      Greengenes should be cited.

433      How many permutations were used for MCPP? This number should be expressed.

438-441           What software/environment was used for these tests?

Results:

87-89   Add numbers and significance for body and waist sizes in some capacity. Referenced figure 1a only shows n=1 and no significance.

87-94   Significance was established by what test? E.g. x and y “were significantly different by ANOVA”

98-101 It is stated that adipose diameters were “significantly” larger. Were diameters tested for significant difference and with what test?

101-102 Again, using what test?

Please explain what test was used for all claims of significance throughout the Results section. Either by outlining which test was used for each data type in the methods or by saying what test was used in your results or figure captions.

193 Methods claim that sequencing was done using Illumina MiSeq amplicon sequencing but statement on this line claims “pyro-sequencing” which is 454. I imagine the authors meant to say “amplicon-based” here (?)

189-217           Again, I think it is important to clarify the time period that this snapshot gut microbiome is representative of as temporal differences have been repeatedly demonstrated in gut microbiota from NAFLD and ALD studies.

194-195           PCoA is a constrained analysis with defined PC axes. As the authors do not evaluate, use, or explain the axes, an unconstrained method such as NMDS is more appropriate. However, the use/misuse of PCoA is a common practice in the field and so is not atypical of the published literature.

200-217           The data show very well that the MCA treated communities have a different microbiome profile than HFD and ND microbiomes. I suggest that the authors do some comparative statistics to establish significance of differences between treatment groups. I thought that this was where the MRPP from the methods would come in but I do not see that reflected in the results (?).

Discussion:

311      Again, reference to pyrosequencing most likely erroneously (?)

320-322           Results do not support authors' claim of maintenance of a normal bacterial population. First, defining a “normal bacterial population” is difficult if not impossible. Second, MCA treated gut microbiota differed from ND gut microbiota if we were to consider ND microbiota "normal". Perhaps rephrase to say something more to the effect of “MCA treatment resulted in populations of important bacteria inferred to be protective or beneficial.”

Conclusions:

447-448           I do not believe that the authors demonstrated maintenance of “gut homeostasis” without temporal data on gut microbiota. Also, communities were different in all treatments, including MCA. I suggest rephrasing this.

450      Again, without temporal data and with differences in all treatment gut microbiota, I find the statement “by maintaining the composition of the gut microbiota” to be erroneous. I suggest rephrasing to something more like “by supporting healthy populations or abundances of inferred important bacteria.”

Author Response

RESPONSES TO REVIWER #1’s COMMENTS:

Reviewer #1:

  1. Comments and Suggestions for Authors

Research Summary:

The authors’ goal was to explore the effects on obesity, metabolic disfunction, and NAFLD when introducing MCA fruit into the diet of mice. They used 24 mice (although indicate 30 initially) in 3 test groups, “Normal Diet”, “High Fat Diet”, and “High Fat Diet with 1% or 3% MCA” although numbers of 1% and 3% mice are not delineated clearly. The author’s looked at liver morphology (sacrifice post-study), TG, TC, HDL-C, GOT, GPT, GIP, and calculated LDL-C levels throughout the study. Human cell culture was additionally used to simulate fatty liver disease with measurements of lipid content. Western blot was used for protein analysis and ELISA used for PPAR-alpha activity. Gut microbiome was performed using 16S amplicon metagenome sequencing with Illumina MiSeq from fecal samples. Sequencing was processed and analyzed using the QIIME2 pipeline. Statistical analysis was performed using standard analysis although the environment and details are not well explained. The author’s conclude that inclusion of MCA with a high fat diet attenuates diet-induced obesity and prevents insulin resistance. They also conclude that there is an MCA induced change in gut microbiota and that gut microbiota homeostasis is maintained through the introduction of MCA to the diet.

 General perception:

The author’s did a lot of admirable and interesting work as part of this study. In particular, the biometrics collected e.g. blood metabolite levels are well thought out. I greatly appreciate the parallel investigation of mice with human cell culture to make the findings from the mouse model more human relevant. The delineation between MCA 1% and MCA 3% is not clear throughout the paper and needs additional clarification. I do feel that some of the work could be better presented and that claims are sometimes questionably represented or supported by the data as outlined below.

  1. Detailed comments and questions on sections:

General:

Double check comma usage throughout.

General proofreading including for things such as proper tense usage (e.g. is vs has and are vs. were 86-57), wording (e.g. “treatment of MCA” vs. “treatment with MCA” 305) and improper sentences (e.g. 166-168).

RESPONSE: We really appreciate reviewer’s astute suggestion. We have edited this paper by an English editor, Dr. Tim J Harrison, to avoid grammatical errors and inappropriate word/ terms. The certification was shown as below.

4 McWalters Fields

St Andrews

Fife KY16 0FH, UK

Dr Cheng Huang

Department of Biotechnology and Laboratory Science in Medicine

National Yang-Ming University, Taipei, Taiwan

21 March 2019

Dear Dr Huang,

Please find below an invoice for editing your manuscript “Momordica cochinchinensis Aril Ameliorates Diet-induced Metabolic Dysfunction and Nonalcoholic Fatty Liver by Modulating Gut Microbiota”

Yours sincerely,

Dr Tim J Harrison

Figures 2a and 4d:

I suggest adding scale bars to the microscopy images.

RESPONSE: Following the reviewer’s suggestion, we added scale bars in Figure 2A and Figure 4D, and replaced the two Figures in the revised manuscript. The scale bar is 100 mm in each figures.

Abstract:

The claim is that the study was conducted to measure the preventative and therapeutic value of MCA. However, MCA was introduced at the beginning of the study so does not illustrate how therapeutic it is when introduced to already presenting high fat diet phenotypes as a therapy. Perhaps rephrase.

RESPONSE: Following the reviewer’s suggestion, we removed “and therapeutic” in Abstract section. (line 31)

Introduction:

No major comments

Materials and Methods:

347-355           It is written that 30 mice were purchased but based on groupings and n of 8, 24 mice were actually used in the study although this number is never expressly communicated. I suggest making this clearer in the writing.

RESPONSE: Thanks for noticing the mistakes. In this study, total 32 mice were used and divided randomly into four groups. 8 mice were used in each groups. We have corrected the number in the revised manuscript. (line 373 and 378)

352-354           Why was no MCA control group established? (ND with MCA 1% and 3%). This would have helped explore MCA gut communities related to MCA.

RESPONSE: We really thank reviewer’s insightful comment. Our main purpose in this study is to observe the effect of MCA under HFD condition. The effect of MCA on microbiota in ND group needs further investigation.

352-354           Nutritional information should be provided for the diets, whether custom prepared for the study, commercial feeds, or through citation.

RESPONSE: We are grateful for the reviewer’s comment. We showed that the nutrients of lyophilized MCA used in this study contains 0.82 ± 0.05 mg of lycopene and 1.78 ± 0.04 mg of β-carotene per gram (line 369-371).

353-354           How many mice had 1% MCA and how many mice had 3% MCA? This delineation should be clearly communicated and those numbers should be your true n values.

RESPONSE: We are grateful for the reviewer’s comment. In this study, total 32 mice were used and divided randomly into four groups. 8 mice were used in each groups. We have corrected the number in the revised manuscript. (line 373 and 378)

414      When were fecal samples collected? This is not outlined anywhere in methods. Was the starting microbial community assessed or just the final community? These factors can have implications for findings.

Please see related recent MDPI publication:

LeBrun, Erick S., Meghali Nighot, Viszwapriya Dharmaprakash, Anand Kumar, Chien-Chi Lo, Patrick SG Chain, and Thomas Y. Ma. "The Gut Microbiome and Alcoholic Liver Disease: Ethanol Consumption Drives Consistent and Reproducible Alteration in Gut Microbiota in Mice." Life 11, no. 1 (2021): 7.

RESPONSE: We are grateful for the reviewer’s comment. The fecal samples were collected 3 days before the end of experimental period. We added this information in Materials and Methods section (line 444).

418-422           PCR protocol (temperatures, times, polymerase used etc.) should be explained, referenced to a kit, or cited so that it is reproducible.

RESPONSE: We are grateful for the reviewer’s comment. The 16S library was performed following the guide for Illumina MiSeq System. We added this information to Materials and Methods section in the revised manuscript as followed. “The 16S library was performed according to the library preparation guide for Illumina MiSeq System (part # 15044223 Rev. B).” (line 446-447).

425      What do you mean when you say that sequences were “replicated” in q2-dada2?

RESPONSE: Thanks for noticing the mistake. It should be “dereplicated”. We corrected it in the revised manuscript. (line 458)

429      Greengenes should be cited.

RESPONSE: We are grateful for the reviewer’s comment. We cited a reference for Greengenes in the revised manuscript (line 462 and 614-616) as followed. “DeSantis TZ, Hugenholtz P, Larsen N, Rojas M, Brodie EL, Keller K, Huber T, Dalevi D, Hu P, Andersen GL: Greengenes, a chimera-checked 16S rRNA gene database and workbench compatible with ARB. Appl Environ Microbiol 2006, 72:5069-5072.”

433      How many permutations were used for MCPP? This number should be expressed.

RESPONSE: Thanks for noticing the mistake. We did not use MCPP in this study. We deleted this sentence in the Materials and Methods section in the revised manuscript.

438-441           What software/environment was used for these tests?

RESPONSE: We are grateful for the reviewer’s comment. We added the software used in the Materials and Methods section in the revised manuscript as followed. (line 471-472) “GraphPad Prism 6.01 software (GraphPad, CA, USA) was used to analyze the experimental data.”

Results:

87-89   Add numbers and significance for body and waist sizes in some capacity. Referenced figure 1a only shows n=1 and no significance.

RESPONSE: We are grateful for the reviewer’s comment. As suggested, since Figure 1a only shows n=1 and no significance, we deleted Figures 1a in the revised manuscript.

87-94   Significance was established by what test? E.g. x and y “were significantly different by ANOVA”

RESPONSE: We are grateful for the reviewer’s comment. As suggested above, since Figure 1a only shows n=1 and no significance, we deleted Figures 1a in the revised manuscript.

98-101 It is stated that adipose diameters were “significantly” larger. Were diameters tested for significant difference and with what test?

RESPONSE: We are grateful for the reviewer’s comment. One-way ANOVA followed by Dunnett’s multiple comparison test was used to compare differences of adipose diameters among groups of samples.

101-102 Again, using what test?

Please explain what test was used for all claims of significance throughout the Results section. Either by outlining which test was used for each data type in the methods or by saying what test was used in your results or figure captions.

RESPONSE: We are grateful for the reviewer’s comment. One-way ANOVA followed by Dunnett’s multiple comparison test was used to compare differences among groups of samples.

193 Methods claim that sequencing was done using Illumina MiSeq amplicon sequencing but statement on this line claims “pyro-sequencing” which is 454. I imagine the authors meant to say “amplicon-based” here (?)

RESPONSE: Thanks for noticing the mistakes. It should be “Illumina sequencing”, not “pyro-sequencing”. We corrected it in the revised manuscript. (line 194 and 323)

189-217           Again, I think it is important to clarify the time period that this snapshot gut microbiome is representative of as temporal differences have been repeatedly demonstrated in gut microbiota from NAFLD and ALD studies.

RESPONSE: We are grateful for the reviewer’s comment. For gut microbiota analysis, the fecal samples were collected 3 days before the end of experimental period. We added this information in Materials and Methods section (line 444).

194-195           PCoA is a constrained analysis with defined PC axes. As the authors do not evaluate, use, or explain the axes, an unconstrained method such as NMDS is more appropriate. However, the use/misuse of PCoA is a common practice in the field and so is not atypical of the published literature.

RESPONSE: We are grateful for the reviewer’s comment. PCoA is one of the common analysis methods for micorbiota. We also used more methods like taxonomic profiling, heatmap and LDA for analysis as shown in Figure 6B, C and D.

200-217           The data show very well that the MCA treated communities have a different microbiome profile than HFD and ND microbiomes. I suggest that the authors do some comparative statistics to establish significance of differences between treatment groups. I thought that this was where the MRPP from the methods would come in but I do not see that reflected in the results (?).

RESPONSE: Thanks for noticing the mistake. We did not use MCPP in this study. We deleted this sentence in the Materials and Methods section in the revised manuscript.

Discussion:

311      Again, reference to pyrosequencing most likely erroneously (?)

RESPONSE: Thanks for noticing the mistakes. It should be “Illumina sequencing”, not “pyro-sequencing”. We corrected it in the revised manuscript. (line 194 and 323)

320-322           Results do not support authors' claim of maintenance of a normal bacterial population. First, defining a “normal bacterial population” is difficult if not impossible. Second, MCA treated gut microbiota differed from ND gut microbiota if we were to consider ND microbiota "normal". Perhaps rephrase to say something more to the effect of “MCA treatment resulted in populations of important bacteria inferred to be protective or beneficial.”

RESPONSE: We are grateful for the reviewer’s comment. As suggested, we rephrase the sentence in the Discussion section in the revised manuscript as followed. (line 334-336) “This observation suggests MCA treatment results n population growth of important bacteria inferred to be protective or beneficial.”

Conclusions:

447-448           I do not believe that the authors demonstrated maintenance of “gut homeostasis” without temporal data on gut microbiota. Also, communities were different in all treatments, including MCA. I suggest rephrasing this.

RESPONSE: We are grateful for the reviewer’s comment. As suggested, the use of “gut homeostasis” is not proper. Therefore, we deleted it in the sentence. (line 482)

450      Again, without temporal data and with differences in all treatment gut microbiota, I find the statement “by maintaining the composition of the gut microbiota” to be erroneous. I suggest rephrasing to something more like “by supporting healthy populations or abundances of inferred important bacteria.”

RESPONSE: We are grateful for the reviewer’s comment. As suggested, we rephrased the sentence to “by supporting healthy populations or abundances of inferred important bacteria.” (line 484)

Reviewer 2 Report

The protective properties of MCA on NAFLD is clearly shown by the authors. the manuscript comes with novel concept of protection by MCA in Fatty liver disease. This study by Dr. Huang and co-authors, investigated MCA effect on NFALD is lacking some clarifications in relation to the microbiome data justifying the alteration. The results are presented well and the study design is good with minor corrections.

  1. Please mention the details about the HFD (Source, Company name, Calorimetric profile etc.)
  2. Please provide Oil-O-Red staining for the control, HFD and MCA treated liver tissue to show the Fatty deposits in the liver.
  3. In the Figure 6, the authors show the microbial diversity and profiling of the microbiota. The PCOA1/PCOA2 doesn't reflect  any change in the HFD and the MCO groups as claimed by the authors in other results. The authors should add the 3% MCA group to show the diversity between the groups, since authors have claim that 3% MCA is more effective than 1% MCA.
  4. The authors claim inflammation but have not shown any inflammatory markers in the liver (neutrophils etc.) to confirm inflammation. If the authors provide the above information  the manuscript is written very well.

Author Response

RESPONSES TO REVIWER #2’s COMMENTS:

Reviewer #2:

Comments and Suggestions for Authors

The protective properties of MCA on NAFLD is clearly shown by the authors. the manuscript comes with novel concept of protection by MCA in Fatty liver disease. This study by Dr. Huang and co-authors, investigated MCA effect on NFALD is lacking some clarifications in relation to the microbiome data justifying the alteration. The results are presented well and the study design is good with minor corrections.

  1. Please mention the details about the HFD (Source, Company name, Calorimetric profile etc.)

RESPONSE: We are grateful for the reviewer’s comment. We added the information about the HFD in the Materials and Methods section in the revised manuscript as followed.(line 386-387). “ND (Cat. 5010) and HFD (Cat. 58V8; 494 kcal per 100g, 45% energy from fat) were purchased from LabDiet Inc. (USA) and TestDiet Inc. (USA), respectively.

  1. Please provide Oil-O-Red staining for the control, HFD and MCA treated liver tissue to show the Fatty deposits in the liver.

RESPONSE: We are grateful for the reviewer’s comment. Oil-Red O staining is used to observe the lipid in tissues. Although we did not provide the images of Oil-Red O staining, the evaluation of hepatic triglyceride (TG) and hepatic cholesterol (TC) was shown in Figure 4B and 4C. These data represent the lipid content in the liver of the experimental groups.

  1. In the Figure 6, the authors show the microbial diversity and profiling of the microbiota. The PCOA1/PCOA2 doesn't reflect any change in the HFD and the MCO groups as claimed by the authors in other results. The authors should add the 3% MCA group to show the diversity between the groups, since authors have claim that 3% MCA is more effective than 1% MCA.

RESPONSE: We are grateful for the reviewer’s comment. PCoA is one of the common analysis methods for microbiota. We also used more methods like taxonomic profiling, heatmap and LDA for evaluating the microbiota ecosystem as shown in Figure 6B, C and D. Although we did not provide the microbiota data of 3% MCA, 1% MCA treatment resulted in populations of important bacteria inferred to be protective or beneficial.

  1. The authors claim inflammation but have not shown any inflammatory markers in the liver (neutrophils etc.) to confirm inflammation. If the authors provide the above information the manuscript is written very well.

RESPONSE: We are grateful for the reviewer’s comment. As suggested, we corrected and rephrase the “inflammation” to “damage” because GOT and GPT are biomarker to represent liver function. (line 139 and 162)

Reviewer 3 Report

MM:

  • details of plant origin needed
  • details of how lycopene and b-carotene was quantified
  •  

Results

  • Fig 2A, 4D and 5C needs scale bars.
  • Fig 1C-D, 2B-G, 3C-G, 4ACEF,  the significant differences are questionable with the average and SEM overlapping. There is also high variability in the MC fed mice and the differences are not convincing.
  • Fig 6B-D too small, hard to see text
  • figure titles should contain full words instead of only abbreviations

Discussion

  • mostly well summarised but had grammatical errors 

Author Response

RESPONSES TO REVIWER #3’s COMMENTS:

Reviewer #3:

Comments and Suggestions for Authors

MM:

  • details of plant origin needed

RESPONSE: We really appreciate reviewer’s astute suggestion. We added more details about the plant origin in Materials and Methods section in the revised manuscript as followed. (line 353-357) “The habitat of Taiwan’s native Momordica cochinchinensis locates in the area of Hualien County and Taitung County in Taiwan. Taitung District Agricultural Research and Executive Station (Taitung County, Taiwan) collected and planted the composite strain of Taiwan’s native Momordica cochinchinensis originated from the wild for research investigation.”

  • details of how lycopene and b-carotene was quantified

 RESPONSE: We really appreciate reviewer’s astute suggestion. We added the details of how lycopene and b-carotene was quantified in the Materials and Methods section in the revised manuscript (line 364-369) as followed. “Momodica cochinchinensis aril was extracted with n-Hexane/Methanol/Acetone. Lycopene and β-carotene standards (Sigma-Aldrich, USA) were prepared with acetone. Samples were analyzed using HPLC to analyze lycopene and β-carotene, respectively. The amount of lycopene was determined by the intensity at the wavelength of 475 nm at the retention time between 9–10 minutes. β-Carotene was determined by the intensity at the wavelength of 450 nm at the retention time of 37.4 ± 2.5 minutes.”

Results

  • Fig 2A, 4D and 5C needs scale bars.

RESPONSE: Following the reviewer’s suggestion, we added scale bars in Figure 2A, 4D and 5C, and replaced the three Figures in the revised manuscript. The scale bar is 100 mm in each figures.

  • Fig 1C-D, 2B-G, 3C-G, 4ACEF, the significant differences are questionable with the average and SEM overlapping. There is also high variability in the MC fed mice and the differences are not convincing.

RESPONSE: We are grateful for the reviewer’s comment. Although the variability existed in the MCA groups, the significant differences were shown after statistical analysis.

  • Fig 6B-D too small, hard to see text

RESPONSE: We are grateful for the reviewer’s comment. We will try to contact with journal editors to enlarge the figures when online published.

  • figure titles should contain full words instead of only abbreviations

RESPONSE: We are grateful for the reviewer’s comment. As suggested, we corrected “OA” to “oleic acid” in the figure 4 title. (line 251)

Discussion

  • mostly well summarised but had grammatical errors 

RESPONSE: We really appreciate reviewer’s astute suggestion. We have edited this paper by an English editor, Dr. Tim J Harrison, to avoid grammatical errors and inappropriate word/ terms. The certification was shown as below.

4 McWalters Fields

St Andrews

Fife KY16 0FH, UK

Dr Cheng Huang

Department of Biotechnology and Laboratory Science in Medicine

National Yang-Ming University, Taipei, Taiwan

21 March 2019

Dear Dr Huang,

Please find below an invoice for editing your manuscript “Momordica cochinchinensis Aril Ameliorates Diet-induced Metabolic Dysfunction and Nonalcoholic Fatty Liver by Modulating Gut Microbiota”

Yours sincerely,

Dr Tim J Harrison

Reviewer 4 Report

Reviewer comments for IJMS-1114260

In this manuscript, the authors report their results regarding momordica cochinchinensis aril ameliorates metabolic dysfunction and nonalcoholic fatty liver in HFD induced mice model. These findings try to demonstrate that Momordica cochinchinensis as a supplementary agent may exert a benefit effect via regulation of gut microbiota. this manuscript is interesting and well-writing, however, there are still some issues left to sort out, details as follows:

  1. The author should mention at first that the preprint version of this manuscript (DOI: 10.21203/rs.2.20237/v1) already exists in Research Square, however, we notice that the first author is different.
  2. Although it’s from a Japanese journal, one previous similar paper titled “Fatty Liver Inhibitory Effect of Freeze-Dried Gac (Momordica cochinchinensis) Aril in Rats Fed a High-Fat Diet” (doi: 10.3136/nskkk.63.44) should be compared properly and cited in the manuscript.
  3. Both in vivo and in vitro study, the positive control group is missing, as a scientific study, an appropriate positive control drug is needed for comparison.
  4. Beta diversity of fecal microbiota was compared; however, the difference of alpha diversity is also an important parameter for evaluating the microbiota ecosystem.
  5. According to the previous data and the reviewer’s experience, if food is free to access, HFD can obviously reduce the food intake by around 20% to 40%. Based on this, a high percentage of the MCA group showed a very high level of food intake as compared to other groups. If
  6. The final body weight of 3% MCA group is higher than 1% MCA group, but other factors, like fat weight, adipocyte diameter, liver weight, are opposite to body weight. Please explain the possible reasons in the discussion portion. If MCA can promote appetite? And if the promoted appetite is associated with gut microbiota change.
  7. Why is the data of gut microbiota in 3% MCA group missing?
  8. Information of manufacturer should be mentioned in line 341, 343, 364, 370, 376, 384, 390, 405, 409, 412, 414 etc.
  9. “Blank” should be added in line 342 and 364.

Author Response

RESPONSES TO REVIWER #4’s COMMENTS:

Reviewer #4:

Comments and Suggestions for Authors

Reviewer comments for IJMS-1114260

In this manuscript, the authors report their results regarding momordica cochinchinensis aril ameliorates metabolic dysfunction and nonalcoholic fatty liver in HFD induced mice model. These findings try to demonstrate that Momordica cochinchinensis as a supplementary agent may exert a benefit effect via regulation of gut microbiota. this manuscript is interesting and well-writing, however, there are still some issues left to sort out, details as follows:

  1. The author should mention at first that the preprint version of this manuscript (DOI: 10.21203/rs.2.20237/v1) already exists in Research Square, however, we notice that the first author is different.

RESPONSE: We are grateful for the reviewer’s comment. We performed more experiments in this submission to IJMS. Therefore, the first author is different.

  1. Although it’s from a Japanese journal, one previous similar paper titled “Fatty Liver Inhibitory Effect of Freeze-Dried Gac (Momordica cochinchinensis) Aril in Rats Fed a High-Fat Diet” (doi: 10.3136/nskkk.63.44) should be compared properly and cited in the manuscript.

RESPONSE: We are grateful for the reviewer’s comment. However, we cannot find this paper in PubMed.

  1. Both in vivo and in vitro study, the positive control group is missing, as a scientific study, an appropriate positive control drug is needed for comparison.

RESPONSE: We are grateful for the reviewer’s comment. There is no FDA approved drug for NAFLD treatment. On the other hand, our aim is to study the effect of special diet on gut microbiota. Therefore, no reference drug was used in this study.

  1. Beta diversity of fecal microbiota was compared; however, the difference of alpha diversity is also an important parameter for evaluating the microbiota ecosystem.

RESPONSE: We are grateful for the reviewer’s comment. PCoA is one of the common analysis methods for microbiota. We also used more methods like taxonomic profiling, heatmap and LDA for evaluating the microbiota ecosystem as shown in Figure 6B, C and D.

  1. According to the previous data and the reviewer’s experience, if food is free to access, HFD can obviously reduce the food intake by around 20% to 40%. Based on this, a high percentage of the MCA group showed a very high level of food intake as compared to other groups.

RESPONSE: We are grateful for the reviewer’s comment. We found that there was no significant difference between HFD group and 1% MCA group and 3% MCA increased food intake as compared to other groups. The detailed mechanism of the high level of food intake when treated with 3% MCA needs further investigation.

  1. The final body weight of 3% MCA group is higher than 1% MCA group, but other factors, like fat weight, adipocyte diameter, liver weight, are opposite to body weight. Please explain the possible reasons in the discussion portion. If MCA can promote appetite? And if the promoted appetite is associated with gut microbiota change.

RESPONSE: We are grateful for the reviewer’s comment. Although the body weight of 3% MCA group is not significantly different from that of 1% MCA group, 3% MCA demonstrates a more pronounced effect regarding other pathophysiological indices, including fat weight, HOMA-IR and liver weight, compared with 1% MCA. We added this comment in Discussion section (line 289-292). It has not been observed that MCA in particular increases appetite.

  1. Why is the data of gut microbiota in 3% MCA group missing?

RESPONSE: We are grateful for the reviewer’s comment. Only 1% MCA mice were analyzed for microbiota, because we observed that even 1% MCA dietary intake efficiently improves the metabolic disorder caused by HFD.

  1. Information of manufacturer should be mentioned in line 341, 343, 364, 370, 376, 384, 390, 405, 409, 412, 414 etc.

RESPONSE: We are grateful for the reviewer’s comment. As suggested, we added more information about manufacturers in the Materials and Methods section in the revised manuscript.

  1. “Blank” should be added in line 342 and 364.

             RESPONSE: We are grateful for the reviewer’s comment. We added blank to the sentences.

Round 2

Reviewer 4 Report

Thank you for the positive revision with patience.